

# Rocket-Induced Lower Ionosphere Disturbances Derived from Measurements of VLF Transmitter Signals

Jingyuan Feng[1], Wei Xu[1,4*], Xudong Gu[1,2,4], Binbin Ni[1,4], Shiwei Wang[1], Bin Li[2,3*], Ze-Jun Hu[2,3], Fang He[2], Xiang-Cai Chen[2,3], and Hong-Qiao Hu[2]

[1] School of Earth and Space Science and Technology, Wuhan University, Wuhan, China

[2] Antarctic Zhongshan Ice and Space Environment National Observation and Research Station, Polar Research Institute of China, Shanghai, China

[3] Center for Space Physics and Astronomy, Polar Research Institute of China, Shanghai, China

[4] Hubei Luojia Laboratory, Wuhan, Hubei, China

Corresponding authors: wei.xu@whu.edu.cn and libin@pric.org.cn

**Key Points:**

(1) We report wave-like perturbations in measurements of VLF transmitter signals triggered by rocket launch events in North America and China

(2) The perturbations caused by three rocket launch events exhibit a common feature of two isolated pulses with the periods of 3-7 minutes

(3) The two-pulse perturbations measured in VLF signals are likely caused by shock acoustic waves and the surface reflection

**Abstract**

Rocket launch can induce large-scale atmospheric disturbances, which were mainly investigated using measurements of total electron content (TEC) in previous studies. In this study, we report the perturbations in Very-Low-Frequency (VLF) transmitter signals triggered by three rocket launch events, which, different from TEC measurements, are directly related to the *D*-region ionosphere. Although the rocket type, launch site, transmitting frequency, and receiver location were different, the perturbations in VLF measurements were similar in all three events. They typically occurred ~4-14 minutes after the liftoff, resulted in an amplitude change of up to 14 dB,



and had a common period of ~3-7 minutes. Moreover, all perturbations consisted of two isolated pulses and this feature is notably different from previous measurements. The VLF amplitude change, in general, increases with the rocket weight and decreases with the distance from the launch site. Given the close correlation between rocket launch and VLF measurements, as well as the similarity between these events, these perturbations were likely caused by the shock acoustic waves generated during rocket launch since both the propagation speed and periods were similar.

**Plain Language Summary**

Very Low Frequency (VLF, 3-30 kHz) waves propagate over long distances by bouncing between the Earth's surface and the *D*-region ionosphere, making them a valuable tool for studying ionospheric changes. With the rapid growth of space exploration, rocket launches have become a significant human-made source of ionospheric disturbances. However, most studies have focused on their effects in the upper ionosphere, leaving the response of the *D*-region largely unexplored. In this study, we reported the perturbations in VLF signals triggered by rocket launches. These perturbations exhibited clear two-stage structure, and had a common period of ~3-7 minutes, matching previously observed shock acoustic waves (SAWs) induced by rocket launches. Our findings demonstrate that VLF signals can effectively detect and monitor ionospheric disturbances induced by rocket launches, providing a new method to study human impacts on near-Earth space and improving our understanding of the lower ionosphere.

**1. Introduction**

The *D*-region ionosphere (60-100 km), as a part of the Mesosphere-Lower-Thermosphere (MLT) region, is a critical layer of the Earth's atmosphere (Wait & Spies, 1964; McRae & Thomson, 2004). As the upper boundary of the Earth-Ionosphere Waveguide, the *D*-region ionosphere controls the propagation of radio waves in the Very Low Frequency (VLF, 3-30 kHz) range, which can travel for long distances with minimal attenuation (3-4 dB/Mm; Wait & Spies, 1964). The electron density of *D*-region ionosphere is highly variant (Thomson et al., 2007) as influenced by various space weather events (Inan et al., 2010). Measurements of VLF transmitter signals have been utilized to investigate the *D*-region ionosphere during solar flares (McRae & Thomson, 2004; Xu et al., 2023b), solar eclipses (Singh et al., 2011; Chakraborty et al., 2016; Xu et al., 2019), and energetic particle precipitation from the Van Allen radiation belts (Rodger et al., 2007; Clilverd et al., 2020), as well as gamma-ray bursts (Fishman & Inan,1998; Cheng et al., 2024). However, due



to the challenges of directly observing this altitude range, this region remains one of the least
explored in the Earth's atmosphere (Clilverd et al., 2009; Inan et al., 2010). The subionospheric
VLF technique remains one of the most reliable methods to investigate the variation and evolution
of *D*-region ionosphere (Cummer et al., 1998; Thomson, 2010).
In addition to the above-mentioned events, the *D*-region is also influenced by atmospheric waves
propagating upward from the lower atmosphere (Haldoupis & Pancheva, 2002; Haldoupis et al.,
2004; Silber & Price, 2017). Gravity waves (GWs) and shock acoustic waves (SAWs) are two
important types, which are often generated by natural events such as volcanic eruptions,
earthquakes, and thunderstorm activity (NaitAmor et al., 2018; Mahmoudian et al., 2021). Besides
natural sources, human activity can also drive large-scale atmospheric disturbances; rocket launch
is well known to be capable of generating impulsive atmospheric SAWs and GWs (Lin et al., 2017;
Chou et al., 2018). To analyze the impacts of rocket launch on the ionosphere, various studies have
utilized the measurements of Total Electron Content (TEC). Mendillo et al. (1975) first
reported the TEC depletion following the launch of Skylab space station, while Lin et al. (2017)
observed Concentric Traveling Ionospheric Disturbances (CTIDs) that were consistent with the
dispersion relation of gravity waves. Moreover, Chou et al. (2018) revealed that gigantic SAWs
were generated during the launch of SpaceX Falcon-9 rocket. More recently, Park et al. (2022)
found plasma density holes lasting for several hours after the rocket launch near Cape Canaveral.
However, these studies were mostly based on TEC measurements, which are more related to the
*F*-region electron density, but how the *D*-region, through which the atmospheric disturbance would
propagate, was influenced by the rocket launch was rarely investigated.
Measurements of VLF transmitter signals are particularly suitable to resolve this problem since
they carry direct information about the *D*-region ionosphere (McRae & Thomson, 2004).
Compared to TEC measurements, the VLF technique can capture rapid and localized disturbances
in the lower ionosphere with high sensitivity and wide spatial coverage (Marshall & Snively, 2014;
Yue & Lyons, 2015) as being widely employed to analyze the ionospheric disturbances caused by
cyclones, thunderstorm, and semi-diurnal tides (NaitAmor et al., 2018; Patil et al., 2024).
NaitAmor et al. (2018) reported measurements of the wave-like feature from VLF transmitter
signals, which was suggested to originate from traveling ionospheric disturbances due to gravity
waves generated by tropical storms and hurricanes. Mahmoudian et al. (2021) found that the VLF
measurements can be utilized to investigate the evolution of the semi-diurnal-tides in various



regions with high spatial resolution. Patil et al. (2024) analyzed the VLF measurements during
the cyclonic storm Fani, and found oscillations with periods of 13-20 minutes from VLF
measurements. Nevertheless, measurements of VLF disturbance caused by rocket launch have not
been previously reported. Saha et al. (2020) have reported VLF disturbance induced by the launch
of Geosynchronous Launch Vehicle rocket from Sriharikota, India on August 27, 2015. The
disturbance occurred 134 seconds after the rocket launch, and the amplitude change was ~3 dB.
In this study, we report the disturbances of VLF transmitter signals induced by three rocket-launch
events in China and North America. Similar wave-like perturbations have been found in all three
events, although the rocket type was different and the VLF signals were emitted by different
transmitters and received by our detectors at different locations. These VLF perturbations were
different from those reported in Saha et al. (2020) and consisted of two isolated pulses with a
common period of 3-7 minutes. We have quantified the periodic oscillation, spatial variations, and
their relation with rocket-induced atmospheric waves. This study provides new insights into how
rocket launch influences the lower ionosphere, and demonstrates the capability of VLF
measurements as an effective tool to monitor atmospheric disturbance.

**2. Instrument and Data**
The data utilized in this study were recorded using the VLF wave detection system developed by
Wuhan University (Zhou et al., 2020; Gu et al., 2022). This system can detect radio waves with
frequencies between 1-50 kHz, with a dynamic range of ~110 dB and a timing accuracy of ~100
ns. Its core components include magnetic loop antennas, an analog front-end, and a digital receiver
module (Gu et al., 2022). Similar to the Atmospheric Weather Electromagnetic System for
Observation, Modeling, and Education (AWESOME) instrument (Cohen et al., 2009), two
triangle-shaped magnetic loop antennas were set up orthogonally in the East-West and North-
South directions. We have established a network consisting of ten receivers in the central area of
China, as well as another receiver at the Great Wall Station (GWS) in Antarctica. VLF data
collected from this network have been widely used in studies of ionospheric and atmospheric
phenomena, such as lightning discharges (Yi et al., 2020; Gu et al., 2021; Xu et al., 2023a), solar
flares (Wang et al., 2024), solar eclipses (Cheng et al., 2023; Gu et al., 2021), and sunrise and
sunset effects (Wang et al., 2020). The amplitude and phase of VLF measurements were calculated



by considering that the transmitter signals were modulated using the Minimum-Shift Keying
(MSK) method. We use the amplitude data for the analysis of rocket launch in this study since the
phase data are well known to be unstable (Thomson et al., 2007; Gross et al., 2018).
In this study, we used the VLF data collected in Shiyan and Suizhou, China, as well as GWS in
Antarctica. For the Shiyan and Suizhou station, we mainly focus on the VLF signals from the
NWC transmitter (19.8 kHz, 21.82°S, 114.12°E). As for the GWS station, the VLF signals from
NLK (24.8 kHz, 48.20°N, 121.92°W) and NML (25.2 kHz, 46.36°N, 98.34°W) were analyzed.
**3. Observational Results**
**3.1. Event #1: August 04, 2022**
At 13:57:00 UT on August 4, 2022, a New Shepard rocket was launched from the Corn Ranch
Spaceport (which is known as Launch Site One, LSO) in Texas. This rocket was single stage and
specifically developed for scientific experiments and space tourism. It was powered by a BE-3PM
engine that utilizes liquid oxygen and hydrogen as the fuel, with a takeoff weight of ~40 tons and
a payload capacity of 5 tons. According to the mission data (available at
https://www.blueorigin.com), this rocket experienced a maximum dynamic pressure about 1
minute after the launch. The engine cutoff occurred at an altitude of ~57 km, approximately 2
minutes and 21 seconds after the liftoff, followed by the separation of the crew capsule and booster.
The capsule reached its peak altitude of ~107 km approximately 1 minute and 42 seconds later.
Around 5 minutes and 35 seconds after the rocket launch, both the booster and the capsule started
descending, eventually landing on the designated pads. The total duration from the liftoff to
touchdown was approximately 10 minutes and 20 seconds.
We have first checked the solar and geomagnetic activity before and after the rocket launch. The
X-ray fluxes remained at a low level ($<10^{-6}$ W/m²), indicating no significant flaring activity. The
solar wind velocity fluctuated between 410-420 km/s, while the Dst and Ap index was 4 nT and
within the range of 4.25 to 5 nT, respectively.
Figure 1a shows the great circle paths from the NLK and NML transmitters to GWS, both of which
are not far away from LSO. Figures 1b and 1c show the amplitude of VLF signals emitted by the
NLK and NML transmitters as received at GWS during the rocket launch, corresponding to the
NLK-GWS and NML-GWS paths, respectively. A significant increase in VLF amplitude was





observed almost at the same moment from both paths, which are close to the launch site.
We first use a third-order polynomial fit in order to estimate the quiet-time trend for the VLF data
collected before and after the event. VLF signals exhibit typical diurnal variation as caused by the
variation of $D$-region ionosphere and this step was conducted in order to isolate the disturbance
caused solely by the rocket launch. The residual disturbances, obtained by subtracting the quiet-
time trend from the raw VLF measurements, are shown in Figures 1d and 1e. For both NLK-GWS
and NML-GWS paths, the disturbance appears to consist of two distinct phases, although the
detailed structure is slightly different.
To extract the perturbation, we have utilized a fifth-order Butterworth bandpass filter. The cutoff
periods were set to be between 2 and 12 minutes, which is chosen based on previous studies on
rocket-induced ionospheric disturbances (e.g., Lin et al., 2017; Liu et al., 2018). The filtered VLF
data are shown in Figures 1f and 1g for the NLK-GWS and NML-GWS paths, respectively. For
the NLK-GWS path, the wave-like perturbations were comprised of a large amplitude perturbation
(~2.82 dB), followed by a subsequent amplitude change of ~1.26 dB. The initial wave reached the
peak at ~14:08:00 UT, and the second wave reached the maximum at ~14:59:10 UT.  As for the
NML-GWS path, the first amplitude change was ~2.80 dB and the second was ~0.46 dB. The
corresponding time of the first and second peak was ~14:07:50 and ~14:39:00 UT, respectively.
Despite the difference in peak time, the overall structure was similar to those measured from the
NLK-GWS path.
Figures 1h and 1i show the wavelet spectra of these perturbations. The black contour marks the
cone of influence, and the blue contour marks the 95% confidence level. For the NLK-GWS path,
the perturbation had typical periods of ~7 and ~5.5 minutes. As for the NML-GWS path, the
periodicity of perturbation was similar (~7 and ~3.5 minutes). For both paths, the periodicity was
close to previously reported values for shock acoustic waves induced by rocket launches (e.g.,
Chou et al., 2018; Xie et al., 2025).





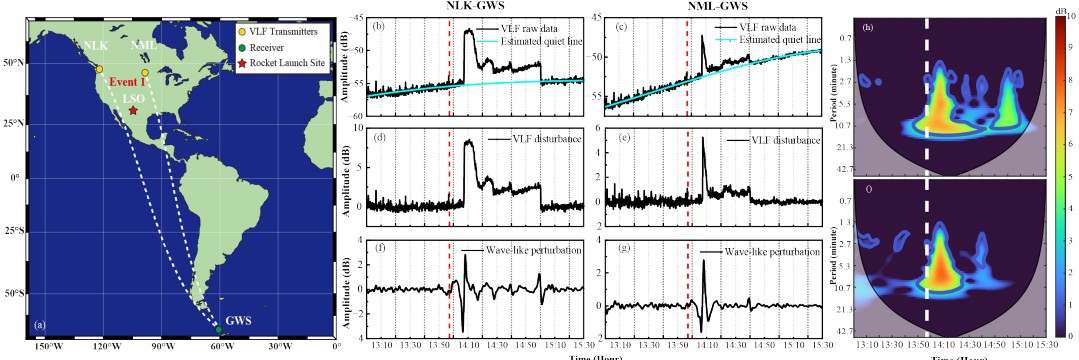


**Figure 1.** (a) Great circle paths from the NML and NML transmitters to our VLF receiver at GWS in Antarctica. The Corn Ranch Spaceport launch site is marked using red star. The VLF data collected during the rocket launch on August 4, 2022 from the NLK-GWS and NML-GWS paths. (b, c) The VLF raw data (black line) and estimated quiet-time curve (cyan line). (d, e) The detrended VLF disturbances. (f, g) The wave-like perturbations. (h, i) Wavelet spectra of the isolated wave-like perturbations. The black curve marks the cone of influence and the blue contours mark the 95% confidence level. The red and white dashed lines mark the rocket launch time.

## 3.2. Event #2: September 20, 2021

On September 20, 2021, the Long March 7 rocket was launched from the Wenchang Spacecraft Launch Site in Hainan, China at ~07:10:00 UT. This was a medium-size and two-stage rocket, which was equipped with four boosters powered by liquid oxygen and kerosene. Approximately 3 minutes and 8 seconds after the liftoff, the first-stage engine was shut down. The second stage was ignited 3 seconds later with a planned burning of 6 minutes and 48 seconds. Throughout this event, the X-ray fluxes were on the order of $10^{-7}$ W/m², the solar wind velocity was 305-308 km/s, the Ap index was lower than 0.5 nT, and the Dst index was approximately −3 nT.

Figure 2a shows the great circle paths from the NWC transmitter to our receivers in Suizhou and Shiyan, China, both of which are close to the Wenchang Spacecraft Launch Site. Figures 2b and 2c show the measurements of VLF signals from the NWC transmitter as received in Shiyan and Suizhou, respectively. It is clear that, for both transmitter-receiver paths, the VLF measurements also consisted of a two-stage decrease in VLF amplitude. For the NWC-Shiyan path (Figure 2f), the amplitude change for the two stages was approximately −10.59 dB and −10.99 dB. The first



perturbation occurred at ~07:24:20 UT, reached its minimum at ~07:26:10 UT, and recovered at
~07:28:10 UT. The second wave occurred at ~07:35:10 UT, reached its minimum at ~07:36:50
UT, and recovered at ~07:38:40 UT. The duration of these two perturbations was 3.8 minutes and
3.5 minutes, respectively. Similar change has been found for the NWC-Suizhou path (Figure 2g).
The maximum reduction of VLF amplitude was −13.44 dB and −13.87 dB. The onset and recovery
time was almost identical to those of the NWC-Shiyan path.
The wavelet analysis results are shown in Figures 2h and 2i. These perturbations have
characteristic periods of approximately 5.3-5.5 minutes, similar to those found for the NLK-GWS
path in event #1. Given the close correlation in space and time between the two paths and rocket
launch, as well as the similarity with event #1, these perturbations are most likely induced by the
Long March 7.

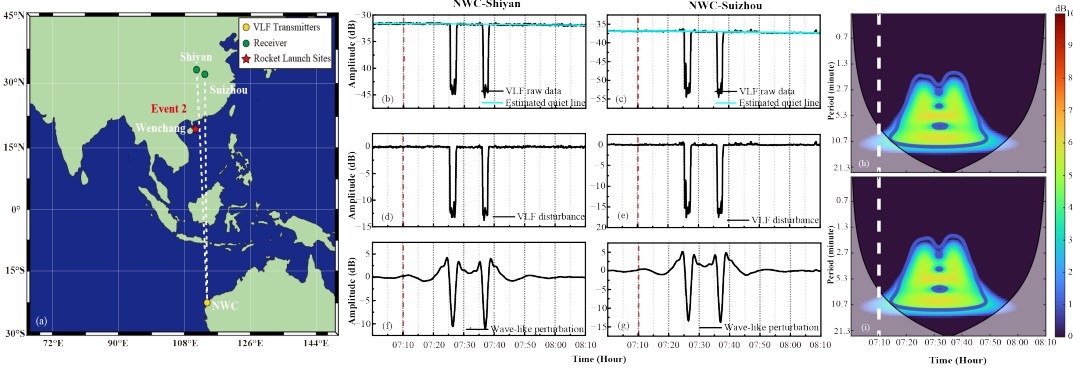


**Figure 2.** Similar to Figure 1, but for the rocket launched from the Wenchang Spacecraft Launch
Site on September 20, 2021. The VLF data emitted by NWC and recorded in Shiyan and Suizhou
were used for this event.
**3.3. Event #3: October 31, 2022**
At 07:37:00 UT on October 31, 2022, the Long March 5B rocket was launched from the Wenchang
Spacecraft Launch Site. This was a one-and-a-half-stage carrier rocket with four boosters powered
by liquid oxygen and kerosene. The boosters were shut down and separated approximately 2
minutes and 53 seconds after the launch, followed by the shutdown of core stage about 8 minutes
and 1 second after the liftoff. During this event, the X-ray fluxes were on the order of $10^{-6}$ W/m$^2$,
the solar wind velocity was ~440 km/s, and the Dst index and Ap index was in the range of 10-15





nT, indicating quiet conditions.
Figure 3 shows similar results with Figure 2, but for the VLF signals emitted from NWC as
measured in Suizhou, China. Note that the Shiyan receiver did not work properly during this event.
Similar to event #1 and #2, these results exhibited a clear two-stage decrease in VLF amplitude.
As shown in Figure 3d, the first perturbation started at ~07:41:30 UT, decreased to the minimum
at ~07:43:10 UT, and recovered at ~07:45:10 UT. The subsequent perturbation started from
~07:45:20 UT, decreased to the minimum at ~07:46:50 UT, and recovered from ~07:48:20 UT.
The duration of these two perturbations was 3.6 and 3 minutes, respectively, both with a typical
period of 3.7 minutes, similar to events #1 and #2.

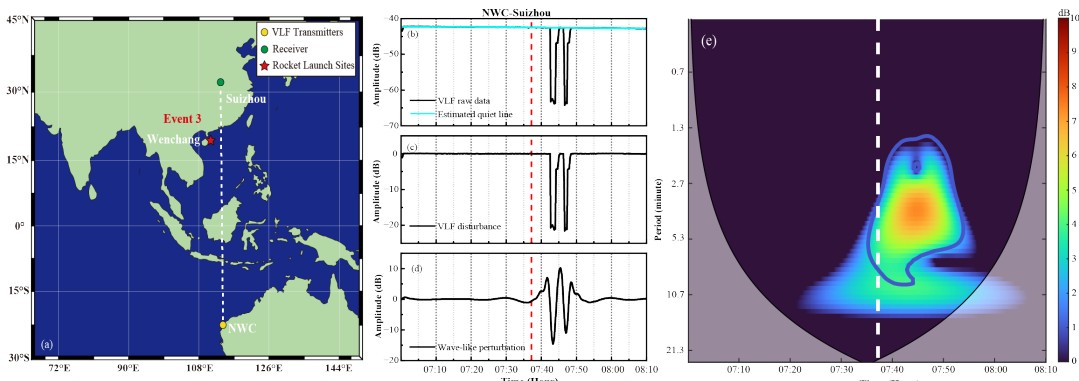


**Figure 3.** Similar to Figure 2, but for the rocket launched on Wenchang Spacecraft Launch Site
on October 31, 2022.

**4. Discussion and Conclusion**
In this study, we reported the VLF disturbances caused by three rocket-launch events between
2021 and 2022. The VLF disturbances during these events are shown in Figures 1-3 and
summarized in Table 1. Table 1 shows the launch sites, VLF propagation paths, distances, the
begin time, the interval between two perturbations, and the periods of VLF disturbance.
For all three events, although the transmitting frequency and receiver sites were different, the
perturbations on VLF signals typically appeared ~4-14 minutes after the rocket lift off and
exhibited a clear two-stage structure. Since similar disturbance has been found from different
events and different paths in the same event, it is believed that these perturbations are more related





with rocket launch, but not due to temporarily shutdown or sudden fault of VLF transmitters. These two-stage perturbations suggest a recurring interaction between rocket-induced atmospheric waves and the *D*-region ionosphere. This pattern is, in general, in line with previous studies: rocket launches have been found to generate traveling ionospheric disturbances (TIDs) and localized ionospheric anomalies (Li et al., 1994; **Bowling** et al., 2013; Ding et al., 2014). However, we emphasize that the VLF disturbance reported in this study consisted of two isolated pulses, which are inherently different in form and structure from the single-peak pulse reported in Saha et al. (2020).

Moreover, all perturbations have been found to have a common period of 3-7 minutes and the speed of these disturbances, if generated by the rocket launch, was in the range of 500-1000 m/s, both of which are consistent with those of shock acoustic waves triggered by rocket launch (e.g., Lin et al., 2017; Chou et al., 2018; Xie et al., 2025). For example, Lin et al. (2017) have found that the V-shaped shock acoustic wave triggered by launch of SpaceX Falcon 9 rocket had a period of ~8-9 minutes. Xie et al. (2025) reported huge ionospheric hole and TIDs during rocket launch in China; the corresponding period of ionospheric disturbances was ~7-8 minutes.

The two-stage perturbation in VLF measurements likely stems from complex interactions between the rocket-induced shock acoustic waves and the *D*-region ionosphere. In this study, the intervals between the two perturbations were found to be 4-50 minutes, which is relatively short and unlikely caused by long-term effects such as the propagation of gravity waves (Chou et al., 2018). Similar phenomena of multi-stage ionospheric perturbation have been reported during rocket launch. Arendt (1971) first reported multiple-stage perturbation induced by the launch of Apollo 14, and the separation time between different stages was approximately 1 hour. The perturbation mainly occurred in the bottom of *F*-region, and was suggested to originate from the acoustic waves triggered by supersonic shock. Li et al. (1994) reported a first pulse and a delayed wave, with an interval of ~12 minutes, during the rocket launch from Cape Canaveral, and attributed the delayed wave to the surface reflection of shuttle-generated disturbance (Lin et al., 2014).

**Table 1.** The launch sites, VLF propagation paths, distances, begin time, amplitude change, interval between two pulses, and the periods of VLF disturbance for events #1-3.




| Mission | Launch Site | Path | Distances (km) | The begin time (UT) | Δ Time (minutes) | Δ Amplitude (dB) | Periods (minutes) |
|---|---|---|---|---|---|---|---|
| Event 1 | LSO | NLK-GWS | ~929.6 | ~ 14:06:20 & ~14:54:40 | 48.33 | ~2.82 & ~1.26 | ~7 & ~5.5 |
| | | NML-GWS | ~1138.5 | ~ 14:06:10 & ~14:36:50 | 30.67 | ~2.80 & ~0.46 | ~7 & ~3.5 |
| Event 2 | Wenchang | NWC-Shiyan | ~120.38 | ~ 07:24:20 & ~07:35:10 | 10.83 | ~−10.59 & ~−10.99 | ~5.5 |
| | | NWC-Suizhou | ~280.73 | ~ 07:24:20 & ~07:35:10 | 10.83 | ~−13.44 & ~−13.87 | ~5.3 |
| Event 3 | Wenchang | NWC-Suizhou | ~280.73 | ~07:41:30 & ~07:45:20 | 3.83 | ~−14.64 & ~−11.04 | ~3.7 |

Despite the similarity, there were some differences in VLF disturbance between the three events: the amplitude change in event #1 was positive, while the changes in events #2 & #3 were negative. This is not unexpected considering that the VLF measurements at a given location is caused by the interference of different propagating waveguide modes, which could increase or decrease depending on the phase difference between different modes (Marshall, 2014; Xu et al., 2021). Moreover, the VLF amplitude change in event #1 was much smaller than those in events #2 and 3 (see Table 1). We speculate that these differences could be related to the difference in the rocket weight and distance. The takeoff weight and payload of New Shepard rocket (event #1) were approximately 40 tons and 5 tons, respectively, while the weight of the rocket in event #2 and #3 was 597 and 849 tons, respectively. The VLF amplitude change observed during these three events roughly increases with the weight of rocket ship and payload. Of note, the NWC-Shiyan and NWC-Suizhou paths were also closer to the rocket launch site, compared to the NLK-GWS and NML-GWS paths in event #1. As such, larger amplitude change observed from the NWC-Shiyan and NWC-Suizhou paths was conceivable.





Different from event #1, the VLF amplitude change during the first and second perturbation was
similar during events #2 and #3 (see Table 1), which are likely caused by sources with equivalent
energy, for instance, the multi-stage propulsion of rockets. The similar amplitude changes in events
#2 and #3 could be explained by the SAWs induced by the rocket liftoff and the propulsion of the
first stage separation. As for event #1, the VLF amplitude change during the second perturbation
was smaller than that of the first perturbation, which could be due to the surface reflection as
suggested by Li et al. (1994) and Lin et al. (2014). This speculation is also consistent with the fact
that the rocket in event #1 was single-stage, while the rockets in event #2 and #3 were two-stage
and one-and-a-half stage.
Furthermore, the difference in the propulsion system can also contribute to the VLF disturbance.
The New Shepard rocket used a hydrogen-oxygen engine, which primarily released $H_2O$, while
the Long March 7 and 5B used liquid oxygen/kerosene, and both $H_2O$ and $CO_2$ were released.
Previous studies have shown that these exhaust products can interact with the $O^+$ ions in the upper
atmosphere, leading to differences in electron depletion (Mendillo et al., 1975; Bernhardt et al.,
1987). Similar effects could be found for the $D$-region ionosphere and need to be better
investigated in future studies.
Although the rocket type, launch site, transmitting frequency, and the receiver location were
different, similar VLF disturbance has been found for three rocket-launch events between 2021
and 2022. Considering the close temporal and spatial correlation between VLF measurements and
rocket launch, as well as the similarity between these events, these perturbations were most likely
caused by SAWs generated during rocket launch. We have carefully checked different sources of
TEC data and cannot find high temporal resolution data for more detailed analysis of these events.
Future studies could use multi-instrument measurements to better understand the two-stage
perturbation caused by rocket launch. Moreover, numerical models can be employed to verify the
specific reason for the two-pulses feature. However, modeling this process and its influence on
VLF signals is very complicated and it requires the combination of atmospheric dynamics and
chemistry models and the VLF propagation model, which is left for the next-step study. This study
demonstrates the capability of VLF technique in remotely sensing the atmospheric disturbance
caused by rocket launch, and highlights the need for network observation of VLF signals, which
can potentially infer the spatial evolution of these disturbances.




## Open Research

The VLF data and codes used to generate the figures reported in this paper can be obtained from https://doi.org/10.5281/zenodo.15531305. Xudong Gu (guxudong@whu.edu.cn) is the data manager and technical engineer for the maintenance of VLF instruments and data processing.

## Acknowledgment

This work was supported by the National Natural Science Foundation of China (Grant Nos. 42025404, 42188101, 42274205, 41874195, 42074119, 42130210), the National Key R&D Program of China (2022YFF0503700), the Antarctic Zhongshan Ice and Space Environment National Observation and Research Station (NO. ZSNORS- 20242603) and the open fund of Hubei Luojia Laboratory (Grant No. 220100051). The authors acknowledge the Chinese Meridian Project II, and the Chinese National Antarctic Research Expedition for the support to the VLF wave detection program at GWS, and the Tencent Xplore prize.

329

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
