# Peer review of "Rocket-Induced Lower Ionosphere Disturbances Derived from Measurements of VLF"

_EGUsphere, 2025_

## Community Comment (CC1)

**31 October 2022. NWC recorded at Seattle (0.1 s resolution).**

[Figure]

**20 September 2021. NWC recorded at Seattle (0.1 s resolution):**

[Figure]

04 August 2022. NLK recorded at Seattle (0.1 s resolution):

[Figure]

NLK to Rothera, Antarctica, (67˚S, 68˚W), 04 August 2022 (5 s resolution):

[Figure]

NML to Rothera, Antarctica, (67˚S, 68˚W), 04 August 2022 (5 s resolution):

---

## Author Comment (AC1)

The authors thank the editor and all reviewers for carefully reading our manuscript and providing valuable comments. The point-to-point responses to the reviewers' comments are provided below. We have also added more details/discussion and revised the text/figures accordingly.

**Reviewer #1**

This paper describes new observations of in the D-region from rocket launches. This paper should be published after considering the follow changes:

1. The authors should start with a statement of the discovery first and then discuss how they are different from previous measurements. Besides VLF waveguide amplitude and phase and radio beacon TEC data, mention should be made of ionosonde observations. We recommend the abstract be rewritten as:

Very-Low-Frequency (VLF) signals from ground transmitters propagating for long distances in the earth-ionosphere waveguide showed changes that were coincident with three rocket launch events. These launches produced acoustic disturbances in the D-region ionosphere. Although the rocket type, launch site, transmitting frequency, and receiver location were different, the perturbations in VLF measurements were similar in all three events. Moreover, all perturbations consisted of two isolated pulses and this feature is notably different from previous measurements. The VLF amplitude change, in general, increases with the rocket weight and decreases with the distance from the launch site. Given the close correlation between rocket launch and VLF measurements, as well as the similarity between these events, these perturbations were likely caused by the shock acoustic waves generated during rocket launch since both the propagation speed and periods were similar.

These events are different than rocket launches that induce (a) large-scale atmospheric disturbances in the F-region, which were mainly investigated using measurements of total electron content (TEC) in previous studies [Mendillo et al., 1975; Bernhardt, 1987] or (b) acoustic wave disturbances that are detected with high frequency sounders of the

ionosphere [Mabie, et al., 2016 or Mabie and Bullett, 2022].

**Reply:** Thank you for the suggestion. We have now revised the abstract to the following:

Lines 23-36: "*Very-Low-Frequency (VLF) signals from ground transmitters propagating for long distances in the earth-ionosphere waveguide exhibited perturbations that were coincident with three rocket launch events. Although the rocket type, launch site, transmitting frequency, and receiver location were different, the perturbations in VLF measurements were similar in all three events. They typically occurred ~4-14 minutes after the liftoff, resulted in an amplitude change of up to 14 dB, and had a common period of ~3-7 minutes. Moreover, all perturbations consisted of two isolated pulses and this feature is notably different from previous measurements. The VLF amplitude change, in general, increases with the rocket weight and decreases with the distance from the launch site. Given the close correlation between rocket launch and VLF measurements, these perturbations were likely caused by the shock acoustic waves generated during rocket launch since both the propagation speed and periods were similar. The perturbations in VLF transmitter signals triggered by three rocket launch events are directly related to the D-region ionosphere, which, different from (a) large-scale atmospheric disturbances in the F-region, mainly investigated using measurements of total electron content (TEC) in previous studies or (b) acoustic wave disturbances that are detected with high frequency sounders of the ionosphere.* "

2. Mention that VLF measurements along long propagation paths in the earth-ionosphere waveguide has also shown impacts of the lunar tide and man's lack of activity on the weekend [Bernhardt, Price, and Crary, 1981].

**Reply:** Thank you for this comment. Indeed, VLF measurements can be influenced by the lunar tide, as well as human activity.

We have made the following changes:

Line 86-94: "*Measurements of VLF transmitter signals are particularly suitable to resolve this problem since they carry direct information about the D-region ionosphere (McRae & Thomson, 2004). Compared to TEC measurements, the VLF technique can capture rapid and localized disturbances in the lower ionosphere with high sensitivity*

*and wide spatial coverage (Marshall & Snively, 2014; Yue & Lyons, 2015) as being widely employed to analyze the ionospheric disturbances caused by cyclones, thunderstorm, semi-diurnal tides, and human activities (Bernhardt et al., 1981; NaitAmor et al., 2018; Patil et al., 2024). Bernhardt et al. (1981) identified lunar tidal oscillations and a weekly variation associated with reduced anthropogenic power line radiation on weekends from phase and amplitude of VLF measurements.*"

3. Please add these references:

Paul A. Bernhardt, Kent M. Price, James H. Crary, Periodic fluctuations in the earth-ionosphere waveguide, Journal of Geophysical Research, Vol. 86, No. A4, pages 2461-2466, April 1, 1981

Mabie, J.; Bullett, T. Multiple Cusp Signatures in Ionograms Associated with Rocket-Induced Infrasonic Waves. Atmosphere 2022, 13, 958. https://doi.org/10.3390/atmos13060958

Mabie, J.; Bullett, T.; Moore, P.; Vieira, G. Identification of rocket-induced acoustic waves in the ionosphere. Geophys. Res. Lett. 2016, 32, 43.

**Reply:** Thank you for this comment. We have added these references to the revised manuscript:

Lines 72-85: "*To analyze the impacts of rocket launch on the ionosphere, various studies have utilized the measurements of Total Electron Content (TEC) and ionosonde. Mendillo et al. (1975) first reported the TEC depletion following the launch of Skylab space station, while Lin et al. (2017) observed Concentric Traveling Ionospheric Disturbances (CTIDs) that were consistent with the dispersion relation of gravity waves. Mabie et al. (2016) identified rocket-induced acoustic waves in ionosphere using ionosonde phase observations. Moreover, Chou et al. (2018) revealed that gigantic SAWs were generated during the launch of SpaceX Falcon-9 rocket. More recently, Park et al. (2022) found plasma density holes lasting for several hours after the rocket launch near Cape Canaveral. Mabie and Bullett (2022) reported the first observation of multiple cusp signatures in ionograms triggered by rocket-induced infrasonic waves, and suggested the close correlation between plasma displacements and ionogram*

*deformations. However, these studies were mostly based on TEC and ionosonde measurements, which are more related to the F-region electron density, but how the D-region, through which the atmospheric disturbance would propagate, was influenced by the rocket launch was rarely investigated..*"

Lines 345-347: "*Bernhardt, P. A., Price, K. M., and Crary, J. H.: Periodic fluctuations in the Earth-ionosphere waveguide, J. Geophys. Res., 86, 2461–2466, https://doi.org/10.1029/JA086iA04p02461, 1981.*"

Lines 431-433: "*Mabie, J., Bullett, T., Moore, P., and Vieira, G.: Identification of rocket-induced acoustic waves in the ionosphere, Geophysical Research Letters, 43, https://doi.org/10.1002/2016GL070820, 2016.*"

Lines 434-435: "*Mabie, J. and Bullett, T.: Multiple Cusp Signatures in Ionograms Associated with Rocket-Induced Infrasonic Waves, Atmosphere, 13, 958, https://doi.org/10.3390/atmos13060958, 2022.*"